# The Influence of Whey Protein Isolate on the Quality Indicators of Acidophilic Ice Cream Based on Liquid Concentrates of Demineralized Whey

**DOI:** 10.3390/foods13010170

**Published:** 2024-01-03

**Authors:** Artur Mykhalevych, Magdalena Buniowska-Olejnik, Galyna Polishchuk, Czesław Puchalski, Anna Kamińska-Dwórznicka, Anna Berthold-Pluta

**Affiliations:** 1Department of Milk and Dairy Products Technology, Educational and Scientific Institute of Food Technologies, National University of Food Technologies, Volodymyrska 68 St., 01033 Kyiv, Ukraine; nuftmilk@i.ua; 2Department of Dairy Technology, Institute of Food Technology and Nutrition, University of Rzeszow, Ćwiklinskiej 2D St., 35-601 Rzeszow, Poland; mbuniowska@ur.edu.pl; 3Department of Bioenergetics, Food Analysis and Microbiology, University of Rzeszow, Ćwiklińskiej 2D, 35-601 Rzeszow, Poland; cpuchalski@ur.edu.pl; 4Department of Food Engineering and Process Management, Institute of Food Sciences, Warsaw University of Life Sciences (WULS-SGGW), Nowoursynowska 159C, 02-776 Warsaw, Poland; anna_kaminska1@sggw.edu.pl; 5Division of Milk Technology, Department of Food Technology and Assessment, Institute of Food Sciences, Warsaw University of Life Sciences—SGGW, Nowoursynowska 159c Street, 02-776 Warsaw, Poland

**Keywords:** liquid whey concentrates, protein, ice cream, microstructure, color, microscopy, cryoscopic temperature, probiotic

## Abstract

The use of liquid whey concentrates in the composition of ice cream, especially in combination with other powdered whey proteins, is limited due to their understudied properties. This article shows the main rheological and thermophysical characteristics of ice cream mixes, as well as color parameters, microstructure, analysis of ice crystals and quality indicators of ice cream during storage. The most significant freezing of free water (*p* ≤ 0.05) was observed in the temperature range from the cryoscopic temperature to −10 °C. The microscopy of experimental ice cream samples based on hydrolyzed whey concentrates indicates the formation of a homogeneous crystalline structure of ice crystals with an average diameter of 13.75–14.75 μm. Microstructural analysis confirms the expediency of using whey protein isolate in ice cream, which ensures uniform distribution of air bubbles in the product and sufficient overrun (71.98–76.55%). The combination of non-hydrolyzed whey concentrate and 3% whey protein isolate provides the highest stability to preserve the purity and color intensity of the ice cream during storage. The produced ice cream can be classified as probiotic (number of *Lactobacillus acidophilus* not lower than 6.2 log CFU/g) and protein-enriched (protein supply from 15.02–18.59%).

## 1. Introduction

The use of whey protein concentrates and isolates in dairy products, in particular ice cream, is becoming increasingly popular due to their functional and technological properties [1,2,3]. On the contrary, the use of liquid whey concentrates in the composition of food products is limited due to their understudied properties. The possibility of using liquid concentrates from whey of various origins in yogurts, cream, sour-milk desserts, sour cream and cheeses has been reported [3,4,5,6,7,8]. Whey ice cream based on liquid concentrates of demineralized whey with an increased solids content (39.61–41.61%, including 3.3% protein), in comparison with traditional ice cream recipes of this type [9], differs in improved quality indicators (original taste, absence of defects in consistency and structure), reduced sugar and lactose content in the case of using hydrolyzed concentrates [10,11,12]. An important aspect of whey ice cream production is the additional possibility of milk whey usage as a by-product of cheese manufacturing. The volume of whey processing in the world is still quite low [13], although in general, there is a trend of increasing interest in whey products [14].

At the previous stage of the research, it was shown that the introduction of protein ingredients into the composition of ice cream based on hydrolyzed whey concentrates significantly increases the thixotropy of ice cream mixes. Whey protein isolate (90%) had the most significant effect on the structural and mechanical properties of whey ice cream mixes among the studied protein additives (soy protein isolate, micellar casein, whey protein concentrate and isolate) [15]. The introduction of whey protein ingredients into the composition of ice cream is appropriate not only for the purpose of expanding the range of protein-enriched products [16,17] but also to prevent excessive freezing of free water in mixes and ice cream during low-temperature processing [18]; to ensure the formation of finely dispersed air bubbles in the thickness of the product during freezing [19,20]; and to provide the product with attractive consumer characteristics by improving resistance to melting, overrun and taste [21,22].

Studies on the effect of whey protein isolate on the quality parameters of ice cream have already reported its ability to increase the viscosity of ice cream mixes and resistance to melting and mask the absence or low content of milk fat in the product [23,24,25]. However, whey proteins could lead to the deterioration of the color of ice cream, especially during storage, and also form a bitter aftertaste, which is related to the specific sensory properties of whey [26,27,28]. Roy et al. [25] reported a reduction in ice cream overrun from 94.9% to 33.9% for increasing ice cream protein content from 4% to 10% due to the use of whey protein isolate. The use of hydrolyzed demineralized whey concentrates could be attractive both from the point of view of reducing the lactose content of the product and preventing the formation of sandy and snowy ice cream consistency [29]. However, monosaccharides in the composition of hydrolyzed whey can indirectly affect the formation of the ice cream structure by lowering the cryoscopic temperature [30]. A decrease in the sugar content in ice cream and the presence of lactose hydrolysis products, monosugars (glucose and galactose), which have a lower molecular weight and, accordingly, a higher concentration [31], are able to decrease the cryoscopic temperature and the amount of frozen free water during freezing and hardening [32]. Such changes can reduce the resistance to melting and overrun of ice cream [33,34]. The use of whey protein isolate in ice cream based on hydrolyzed whey concentrates could slow down the freezing process of free water due to its additional binding by whey proteins [35]. It was reported that due to the specificity of the amino acid composition, whey protein isolate belongs to the group of protein additives capable of inhibiting ice recrystallization [36,37,38]. Whey protein isolate could also be used as a functional and technological ingredient (emulsifier, thickener, gelling agent, foaming agent and water-binding agent) in the production of products with characteristics similar to those of classical formulations [39]. In addition, scientists have reported that whey protein isolate is able to support the activity of probiotic bacteria srtains *Lactobacillus* or *Bifidobacterium* in dairy products [40,41], which is an important aspect of this study.

Available scientific information on the influence of whey protein isolate on ice cream quality is characterized by certain contradictions, which are explained by the different degrees of its purification, the quality of the input raw materials and the chemical composition and technology of the selected type of ice cream [19,20,21,23,24,25,27,39].

To our knowledge, there are no publications on the use of whey protein isolate in the formulation of whey ice cream based on liquid concentrates of demineralized whey. The use of liquid hydrolyzed concentrates of demineralized whey in ice cream allows reducing the amount of sugar to 9% and lactose to 1% [9], which could be attractive to individual groups of consumers.

Thus, ***the purpose*** of this research was to study the influence of whey protein isolate on the physicochemical and sensory parameters of whey ice cream.

The following ***tasks*** of the research work were formulated:-To study the influence of whey protein isolate on the physicochemical and rheological properties of mixes and ice cream;-To compare the dynamics of changes in the freezing process in ice cream with whey protein isolate of free water during freezing and subsequent storage of the product at sub-zero temperatures;-To study the microstructure of soft ice cream and determine the main physical and chemical parameters;-To measure the microbiological indicators of ice cream during storage.

## 2. Materials and Methods

### 2.1. Raw Materials

Liquid concentrates of demineralized whey with a solids content of 40% were used to make experimental ice cream samples. Non-hydrolyzed concentrates of demineralized whey were produced by the reconstitution of whey powder with a degree of demineralization of 90% (Herkules, MLEKOVITA, High Mazovia, Poland) in water. Hydrolyzed concentrates of demineralized whey were obtained by lactose fermenting using the enzyme lactase (β-galactosidase) with an activity of 5200 NLU/g (SEROWAR s.c., Szczecin, Poland) and the starter preparation nu-trish^®^ LA-5^®^ containing L. acidophilus (Chr. Hansen A/S, Hoersholm, Denmark). Whey protein isolate 90% (SPOMLEK, Radzyń Podlaski, Poland) was used as a protein supplement. Water, white (regular) sugar, vanillin, activated starter and the stabilization system Cremodan SI 320 (Danisco A/S, Brabrand, Denmark) were used to prepare ice cream mixes. The content of whey protein isolate was explained by its influence on the thixotropic properties of ice cream mixes, which was explained at the previous stage of the experiment [15]. In ice cream based on non-hydrolyzed whey protein concentrate, its maximum effective amount is 3%, while in ice cream based on hydrolyzed whey concentrate, it is up to 5%. The difference in sugar content (11% and 9%) for ice cream samples with non-hydrolyzed and hydrolyzed concentrates of demineralized whey, respectively, is explained by the different degrees of sweetness, which increases during the hydrolysis of lactose into monosaccharides and, accordingly, allows reducing the amount of sugar to 9% in the case of using hydrolyzed concentrates [9]. The content of the stabilization system (0.6%) was chosen in accordance with the manufacturer’s recommendations for the production of low-fat types of ice cream. Liquid concentrates of demineralized whey were made on the basis of water and demineralized whey powder 90% (Herkules, MLEKOVITA, High Mazovia, Poland). Enzymolysis of lactose in concentrates was carried out using the enzyme lactase (β-galactosidase) with an activity of 5200 NLU/g (SEROWAR s.c., Szczecin, Poland) and the starter nu-trish^®^ LA-5^®^ (Chr. Hansen A/S, Hoersholm, Denmark).

### 2.2. Ice Cream Production

#### 2.2.1. Activated Starter

To obtain an activated starter, ultra-pasteurized skimmed milk was heated to an inoculation temperature of 38–42 °C, after which a pre-calculated amount of starter was added. Fermentation was carried out until pH 5.4–5.2.

#### 2.2.2. Hydrolyzed Concentrates

Demineralized whey powder was reconstituted in water at a temperature of (40–45) °C to obtain concentrates with a solids content of 40%. The concentrates were filtered, pasteurized at (85–88) °C for 3–5 min and cooled to the storage temperature for non-hydrolyzed concentrates.

Hydrolyzed concentrates of demineralized whey were obtained according to the technology of Osmak et al. [42]. For the production of hydrolyzed whey concentrates, after pasteurization, they were cooled to (40–43) °C and simultaneously fermented with β-galactosidase preparation and starter nu-trish^®^ LA-5^®^. With the simultaneous introduction of the β-galactosidase enzyme and the starter during the lag phase of L. acidophilus development (2–4 h), the enzyme has time to reveal hydrolytic activity at pH ≥ 5.7, which makes it possible to achieve maximum hydrolysis of lactose within 8 h of enzymolysis.

#### 2.2.3. Ice Cream

Dry components (white sugar, stabilization system, vanilla and whey protein isolate) were mixed and added to preheated water (40–45 °C). Then, liquid concentrates of demineralized whey were added. The obtained mixes were filtered, pasteurized at 83–87 °C for 5 min and homogenized under a pressure of 12.0 ± 2.5 MPa using a laboratory homogenizer-disperser 15M-8TA “Lab Homogenizer & Sub-Micron Disperser” (GAULIN. CORPORATION, Boston, MA, USA). The homogenized mixes were cooled to 38–42 °C, and 3% activated starter was added. The fermentation process was carried out until pH 5.25–5.1, followed by cooling to 2–6 °C and maturation for 24 h. The matured mixes were frozen on a periodic freezer FPM-3.5/380-50 “Elbrus-400” (JSC “ROSS”, Kharkiv, Ukraine). At the first stage of freezing, the mixes were cooled in a cooling cylinder (volume—7 L) to −1 °C at a rotation frequency of the scraper stirrer of 4.5 s^−1^ for 120 s. At the second stage, the mixes were frozen at a rotation frequency of 9 s^−1^ for 180 s to −5.0 ± 0.5 °C. The formulations of experimental samples of ice cream are given in Table 1.

### 2.3. Methods

#### 2.3.1. Chemical Composition

The total solids content in ice cream samples was determined by the arbitration method, the principle of which consists in drying the sample diluted with distilled water and mixed with sand at 102 °C to a constant mass, followed by weighing to determine the mass of the residue.

The protein content was determined using the Kjeldahl method, the fat content using the modified Gerber method [43], and the carbohydrate content using the Bertrand method [44]. The lactose content in whey concentrates and test samples of ice cream mixes was determined using iodometric and refractometric methods [45]. The degree of lactose hydrolysis was calculated based on the found lactose content [46]:Degree of lactose hydrolysis (%) = 100% − residual lactose content (%)(1)

The level of protein supply (%) in the finished product was calculated as the ratio of the protein content to the sum of the protein, fat and carbohydrate content, multiplied by 100.

#### 2.3.2. Rheological and Thermophysical Characteristics of Ice Cream Mixes

After cooling to 2–6 °C, the ice cream mixes were whipped for 5, 10 and 15 min with 5 min breaks according to the modified method of Lim et al. [47]. Foam overrun was determined as the ratio of the volume of the whipped mix to its initial volume, expressed as a percentage. The foam resistance of ice cream mixes was determined according to the modified method of Philips L., according to which a container with a hole in the bottom was used for the foam to drain after whipping [47]. The time during which 50% of the initial volume of the mix used for whipping is formed as a result of foam destruction was taken as an indicator of foam stability. The viscosity of ice cream mixes was determined from warmed samples using a viscometer IKA ROTAVISC lo-vi Complete (IKA, Staufen, Germany). A spindle SP-4 was used to measure viscosity, which was immersed in a prepared sample at 20 °C and a shear rate of 200 rpm. Viscosity values were read after 2 min [48]. Water activity was determined on a water activity analyzer “HygroLab 2” (Rotronic, Bassersdorf, Switzerland) at 20 °C in a measurement range of 0–1 Aw (0–100% rh) [49]. The cryoscopic temperature was set using a cryostat and Beckman thermometer (TL-1). Based on Raoult’s law for non-dissociated molecular solutions, the amount of frozen water at different temperature stages was calculated using the following formula [50]:ω = (1 − t_cr_/t) × 100(2)
where ω is content of frozen water, %; t_cr_ is cryoscopic temperature, °C; and t is the temperature at each stage of technological processing, °C.

#### 2.3.3. Color

Color parameters were determined using a colorimeter (Precision Colorimeter, Model NR 145, Shenzhen, China) and the CIE LAB system. The following parameters were defined: L* as whiteness (from 0—black to 100—white), a* as a color from red (+) to green (−), b* as a color from yellow (+) to blue (−), C* as the purity and intensity of color from gray (C* = 0) to the direction of pure colors (C* = 100) and h° as a color shade (within 0–360°). Before measurement, the device was calibrated against a white standard.

#### 2.3.4. Microstructure

The ice cream sample was taken from the center of the sample in at least three different places and at a distance of 3 cm from the surface of the product, placed at 19 ± 1 °C in a Goryaev chamber covered with glass and immediately subjected to microscopy at a magnification of 160 times. At the same time, the ice crystals melted, but the foam remained, because under these conditions, the air bubble envelopes did not dehydrate. Photomicrographs were obtained using an Olympus CX41 light microscope (Olympus Corporation, Tokyo, Japan) and camera [19].

#### 2.3.5. Analysis of Ice Crystals

Ice cream samples were taken from the center of the sample, from at least 3 different locations, at least 3 cm from the surface of the ice cream, and placed on a glass slide using a spatula and then covered with a glass. The samples were transferred to a microscope with a cooling system (Linkam LTS420, Tokyo, Japan). The recrystallization process was analyzed based on images of ice crystals taken after preparation, using an Olympus BX53 microscope (Olympus Corporation, Tokyo, Japan) with a Linkam LTS420 (Tokyo, Japan) cooling system (temperature range from −196 °C to −420 °C) and an Olympus SC50 camera (Olympus Corporation, Tokyo, Japan). The resulting images were then analyzed using software NIS Elements D (version 5.30.00, Nikon, Tokyo, Japan). Between 300 and 500 crystals were labeled for each sample, and area, equivalent diameter and standard deviation were then calculated using software NIS Elements D Imaging (version 5.30.00, Nikon). Crystal size frequency distributions were calculated using Microsoft Excel 2011 macro data analysis. The relative frequency of any class interval was calculated as the number of crystals in that class (class frequency) divided by the total number of crystals and expressed as a percentage. The X50 parameter was analyzed as the average diameter (DA) for 50% of the crystals in the sample. The mean diameter (DA) and standard deviation (SD) of each class were also calculated. The method of determination is given in scientific works [51,52,53].

#### 2.3.6. Quality Indicators of Ice Cream

The overrun of the ice cream was determined using the weighing method, according to which the ice cream mix was weighed before freezing and the same volume of soft ice cream after freezing. Overrun (O, %) was calculated according to the following formula [54]:O = (M_1_ − M_2_/M_2_) × 100(3)
where M_1_ is the mass of the glass with the mix, g, and M_2_ is the mass of a glass with ice cream, g.

Resistance to melting (the time of the first drop flowing out and the time of accumulation of 10 cm^3^) was determined at an ambient temperature of 22 °C. The ice cream samples were placed on a grid (d = 95 mm, holes 5 × 5 mm, wire thickness 0.5 mm). Then, the time until complete melting was recorded for ice cream melting [51]. Hardness of ice cream samples was determined using a Brookfield CT-3 model texture analyzer. Hardness measurements were made between −6 °C and −2 °C, using texture software Pro CT V1.6 (Brookfield Engineering Laboratories Onc., ABD, Middleboro, MA, USA). A conical probe TA 15/1000 was used for analysis. Test speed was 2 mm/s, distance 15 mm, trigger load 6.8 g, length 40 mm and diameter 60 mm.

#### 2.3.7. Microbiological Analysis

For the study, 5 g of each sample was diluted in 45 cm^3^ of physiological saline. Microbiological analysis was performed under the following conditions: the total number of microorganisms on PCA (Plate Count Agar, Oxoid, Basingstoke, UK) at 30 °C for 48–72 h and the number of intestinal bacteria on VRBL agar (Violet Red Bile agar with lactose, Oxoid, Basingstoke, UK) at 37 °C for 24–48 h. Acidophilic bacilli counts on MRS (De Man Rogosa Sharpe, Oxoid, Basingstoke, UK) and MSE (Mayeux, Sandine & Elliker, Oxoid, Basingstoke, UK) were analyzed at 37 °C for 48–72 h and microscopic fungi and yeast on MEA (malt extract agar, Oxoid, Basingstoke, UK) at 25 °C for 5 days. For Lactobacillus acidophilus, the study was carried out using the plate method with MRS (Biocorp, Warsaw, Poland). Lactobacillus acidophilus was cultivated in microaerophilic conditions with 5% CO_2_ at 37 °C.

#### 2.3.8. Statistical Processing

Analysis of variance (ANOVA) was performed using STATISTICA 13 software. The significance of the test was set at α = 0.05. Data are expressed as mean with standard deviations (±SD), and differences between groups were assessed using Tukey’s HSD test. The study of physicochemical indicators of ice cream samples was conducted three times to ensure the reliability of the data obtained.

## 3. Results and Discussion

### 3.1. Chemical Composition

The chemical composition of liquid concentrates of demineralized whey (Table 2) differs slightly from the examples of liquid whey concentrates given in the literature, which usually have a higher protein content (5.09–11.93%) but a lower solids content (14.09–26.45%), while the content of fat varies from 0.43–0.78% for low-fat concentrates to 6.43–7.82% for full-fat concentrates [3,5,55]. The use of highly demineralized whey to obtain concentrates in this study significantly increases the lactose content to 30.5%, which could significantly affect the quality of ice cream during storage. The hydrolysis of lactose in whey concentrates allows reducing its content to 0.98%, which is lower than the value (4.95%) reported in another study [3]. At the same time, the high proportion of solids (39.92–40.01%) in these liquid concentrates could help to ensure the recommended level of solids in ice cream in the range of 25–35% [42], which is especially important for low-fat types of ice cream. The difference in the obtained values could be explained by the use of whey of different origins, as well as the use of special technologies for the production of concentrates.

Ice cream based on liquid whey concentrates in terms of solids content could be attributed to the full-fat analog (12–18% fat), namely the category of super-premium type of ice cream (40–42% solids) for all samples, except for NHC, which belongs to the premium category (38–40% solids) [56]. The content of high-value whey proteins in ice cream based on whey concentrates exceeds the average protein content (2.6–4.6%) in traditional types of ice cream (10–16% fat) [57]. However, it is insufficient for NHC and HC samples for their classification as ice cream enriched with protein (more than 12% protein supply according to EU regulation No. 1924/2006). Samples of ice cream with whey protein isolate are protein-enriched products, but their protein content does not allow them to be classified as a product source of protein (more than 20% protein supply in accordance with EU Regulation No. 1924/2006), which is a perspective for further research. In general, the data from Table 1 indicate a significant increase in the nutritional value of the developed ice cream compositions based on whey concentrates. However, in terms of the lactose content, the sample NH3% significantly exceeds the others, which could cause defects in consistency and taste during low-temperature storage of the product.

### 3.2. Physical Characteristics of Ice Cream Mixes

Table 3 shows the thermophysical characteristics of ice cream mixes based on liquid concentrates of demineralized whey. The addition of whey protein isolate (3–5%) increases foam overrun and foam resistance, and the maximum effect is observed in ice cream mixes based on hydrolyzed whey concentrates. The foam overrun of the mixes increases after 5 min of whipping and decreases after 10 min, except for samples with hydrolyzed whey concentrate and whey protein isolate, which could be explained by the lower viscosity of the mixes due to the presence of lactose hydrolysis products [58,59], as well as a slightly lower sugar content, which usually appears in ice cream mixes as a component that increases viscosity [56]. Lee and Duggan [60] found that the foam overrun of native mixes with whey protein isolate is significantly higher than microgels with WPI, but their foam stability is slightly lower. In this study, we observed a significant increase in both the foam overrun and foam resistance of whey ice cream mixes due to the fact that the ice cream mix is a multicomponent system in which the synergistic interactions of the ingredients could significantly influence its rheological properties. The combination of whey proteins with monosaccharides provides a significant increase in the foaming properties of mixes. Puangmanee et al. [61] compared the influence of ordinary whey protein isolate and its glycated species with various monosaccharides on the rheological properties of mixes. It was shown that the presence of d-glucose, d-fructose, d-allose and d-psicose additionally increases the foam resistance and foam overrun of the mixes, and this value only increases during the whipping interval from 15 to 30 min. This suggests that the presence of high amounts of monosaccharides in hydrolyzed whey concentrate ice cream mixes exhibits a synergistic effect with WPI. The obtained viscosity values are correlated with the indicators of foam overrun and foam resistance, confirming the regularity of the decrease in this indicator in mixes with a high content of monosaccharides (glucose, galactose). Akalin et al. [62] reported that the addition of 4% WPI resulted in the excessive thickening and gelation of ice cream with an increase in the viscosity of the ice cream mix. Another study reported that the combination of WPI with polysaccharides significantly increased the viscosity of ice cream mixes, which overall had a positive effect on its quality [23].

The cryoscopic temperature of ice cream mixes based on non-hydrolyzed concentrate is slightly higher than samples based on hydrolyzed concentrate, which is explained by the high content of monosaccharides in the latter, as a result of which a depression of the freezing point of ice cream is observed [63]. The high content of solids in all ice cream samples reduces the range of cryoscopic temperature depressions of ice cream mixes within the range of values from −2.39 to −2.95 °C, which is similar to the results of other scientists who studied low-fat or high-protein ice cream [19,64,65,66]. The study of osmotic pressure in the aqueous phase of whey ice cream mixes shows (Table 3) a decrease in water activity in mixes based on hydrolyzed whey concentrate. The addition of WPI reduces water activity, but the most significant effect on this indicator, in our opinion, is the presence of monosaccharides, which have a higher ability to bind free water than sucrose [67]. Thus, samples NHC, NH3%, HC and H3% could be attributed to systems with high water activity (aw = 1.0–0.9), while H5% could be attributed to a food system with intermediate moisture (aw = 0.9–0.6) [68]. A decrease in water activity will affect the quality of ice cream due to a decrease in the freezing point caused by the hydrolysis of lactose in the samples HC, H3% and H5%. This and other influences will be considered in the following sections of this article.

Water in ice cream is in a bound and free state [69]. The latter could freeze in the form of ice crystals at temperatures below the cryoscopic temperature [70]. During product storage, even with slight temperature fluctuations, migratory recrystallization of ice crystals and their fusion occur, which is accompanied by the disappearance of small crystals and the growth of large ones [71]. The recommended storage temperature of ice cream is not lower than −16 to −25 °C [19]; therefore, even with slight fluctuations in it, in ice cream with a high content of free water, the appearance of a coarse crystalline structure is possible. The change in the physical state of the aqueous phase of ice cream has been studied by many scientists [72,73,74], but the use of protein ingredients in its composition requires additional research. Proteins, as well as mono- and disaccharides, could, to some extent, affect the cryoscopic temperature of mixes and, accordingly, the proportion of frozen water and the structure of ice cream. Based on the values of the cryoscopic temperature of the studied mixes (Table 3), the content of frozen water in ice cream was calculated in the temperature range of technological processing from −5 to −40 °C, and it was shown that the proportion of frozen water reached values at the beginning and at the end of freezing from 42.40–52.20% to 92.80–94.03%, respectively (Figure 1).

The maximum difference in the amount of frozen water in the studied samples (up to 9.8%) was found at −5 °C, but with a further decrease in temperature, the difference in the amount of frozen water in ice cream decreased to 1.4%. In general, the most significant freezing of free water was observed in the temperature range from the cryoscopic temperature to −10 °C. Under these conditions, 70.5–71.2% of water froze in the samples NHC and NH3% and 72.9–76.1% in the samples HC, H3% and H5%, which indicates the most significant cryoprotective activity of monosaccharides in the hydrolyzed whey concentrate, including combination with whey protein concentrate. A significant part of the water (up to 17.1–19.7%) continued to freeze when the mixes reached a temperature of −30 °C. The further change in this indicator in the temperature range from −30 to −40 °C was quite insignificant. The obtained data confirm the dynamics of freezing of water in ice cream based on whey concentrates, typical for classic types of ice cream, at low temperatures during the freezing of mixes and hardening and storage of the studied samples.

### 3.3. Color

The lowest value of whiteness L* (*p* ≤ 0.05) was recorded for H5% (Table 4) which contained the largest amount of added whey protein isolate (5%), which gives the product a yellow color [24]. Barros et al. [75] also reported a decrease in whiteness in ice cream samples from 87.65 to 82.45 units depending on the content of concentrated whey (70–280 g per ice cream sample). The obtained L* values for all ice cream samples are lower than the reported results [76], which is explained by the higher content of whey ingredients than in the recipes of known analogs, as well as the absence in these samples of pasteurized or dry milk as raw components that give whiteness to dairy products due to the presence of colloidal particles, such as milk fat globules and casein micelles, capable of scattering light in the visible spectrum [77]. During 14 days of storage, the value of whiteness (L*) in ice cream decreases; however, for samples with hydrolyzed whey concentrate, this process occurs less intensively, which is associated with the effect of lactose hydrolysis, which increases the whiteness of the food system [78]. The values of parameters a* and b* indicate the predominance of green and yellow colors in the product, respectively, which is due to the color characteristics of whey and its processing products [79,80], which were the main ingredients in ice cream production for this study.

A decrease in the parameter b* was observed for HC which means a decrease in the intensity of the yellow color and correlates with the L* indicator, which, on the contrary, increases for this ice cream sample. The addition of whey protein isolate to samples based on hydrolyzed concentrate leads to an increase in b* (increased yellowness). Barros et al. [75] also reported an increase in the b* indicator from 16.27 to 18.02% with an increase in the content of concentrated whey in the composition of ice cream. However, Meneses et al. [80] showed that the b* indicator increased from 57.64 to 60.92 units when adding whey at different mass fractions. A significant difference in the results obtained in this study may be related to the hydrolysis of lactose, as well as the type of whey used. During storage, the degree of green and yellow coloring increases (*p* ≤ 0.05) for all samples. Color intensity and purity (C*) increase for HC, but further addition of whey protein isolate (H3% and H5%) shifts this parameter toward gray colors (less pure and intense color). During storage, the largest decrease in the C* index occurs for NHC, which emphasizes the instability of its color at low temperatures. The NH3% sample shows greater stability to maintain purity and color intensity. The value of the hue (h°) by the position in the spectrum makes it possible to conclude that the ice cream samples are between yellow and green with a predominance toward the first, which is correlated with indicators a* and b*.

### 3.4. Microstructure

Analysis of the microstructure indicates that whey protein isolate (3–5%) contributes to the increase in the size of air bubbles in ice cream, as well as non-hydrolyzed whey concentrate as an ice cream base (Figure 2). The sample HC is characterized by a finer dispersion (average diameter of air bubbles—6.4 μm) and uniform distribution of the air phase in the thickness of the product, compared to NHC (average diameter of air bubbles—8.8 μm), which is due to the higher content of solids in the latter, as well as slightly higher viscosity as a result of significant lactose content. At the same time, the addition of 3% whey protein isolate in combination with non-hydrolyzed concentrate (NH3%) leads to the formation of large air bubbles (up to 12.5–24.1 μm in diameter) due to the increased viscosity of the ice cream mix, which makes it difficult to saturate mixes with air during the freezing process. Whey protein isolate in the samples NH3% and NH5% also leads to the formation of larger air bubbles than in the control samples (NHC, HC); however, their number is significantly lower than for NH3%. In addition, for the above-mentioned samples, a significantly larger number of finely dispersed air bubbles are observed, which show the ability to aggregate among themselves and concentrate around larger air bubbles.

The effect of sticking air bubbles and their uniform distribution in ice cream was observed by scientists when using polysaccharides, protein concentrates and isolates [64,72,81]. In whey ice cream, the size of air bubbles depends not only on the type of raw materials and components but also on the solid content. The small number of solids in MSNF (milk solids non-fat) leads to a structure with inclusions of large air cells [82] and, accordingly, a decrease in the melting rate [25]. The presence of monosaccharides in samples HC, H3% and H5% indirectly affects the distribution of the air phase in the product thickness [83], primarily due to a decrease in the viscosity of ice cream mixes, which increases their saturation by air during freezing. El-Hadad et al. [84] reported that emulsifiers and stabilizers can also affect the even distribution of air bubbles in ice cream. At the same time, the homogenization under pressure of ice cream mixes could have an indirect effect [65], since finely dispersed fat takes an active part in the stabilization of air bubbles [85]. In low-fat ice cream, as in our case, the stabilizing role is performed by whey proteins [86], the high content of which forms a coagulation-type structure, which subsequently leads to a high overrun of ice cream after freezing.

### 3.5. Microscopy Analysis

The recommended size of ice crystals in ice cream should not exceed 50 μm [87]; however, the formation of ice crystals with a size of 10 to 20 μm, according to some scientists, gives ice cream the proper smoothness and creaminess [53,88], while ice crystals larger than 50 μm give the product an undesirable texture [89,90]. Crystal diameters smaller than 20 μm were found in ice cream after 24 h in the freezer storage and may contribute to the stabilization of the crystal structure during longer storage of the product, which needs to be verified experimentally in further studies. Table 5 clearly demonstrates changes in the equivalent diameter of ice crystals for the studied samples. Samples H3% and H5% are characterized by crystals with the smallest sizes at the level of 12.23–13.18 μm (Table 5, Figure 3), which corresponds to the average diameter of ice crystals in milk ice cream stabilized with special cryoprotectants [53].

In ice cream samples (HC, H3% and H5%), 50% of the tested crystal diameters (parameter X50) did not exceed 12.23–13.68 μm (Figure 3). This crystal size after 24 h of ice cream production will ensure stable storage of this product at the specified temperature (−18 °C). Even if temperature fluctuations occur and crystals grow as a result of the recrystallization process, they will not exceed the recommended size (25–50 µm). It has been reported that the growth of ice crystals to diameters greater than 50 μm after one month of storage in dairy ice cream is possible if stabilizers are not used [51,91,92]. Analyzing the diameter of ice crystals according to the studied parameter X50, it is clearly seen that for the samples NHC and NH3%, the largest ice crystals were formed at the level of 16.43–25.96 μm (Table 5, Figure 3). Whey protein isolate contributes to the formation of a structure with increased water-holding capacity, which leads to the formation of a strong three-dimensional network in the NH3%, HC, H3% and H5%. The resulting structure can even mechanically counteract the growth of ice crystals.

Scientists have proven that adding proteins to ice cream can lead to the growth of non-hexagonal crystals, the structure of which is more favorable for binding numerous water molecules [93]. It was also shown that the shape of ice crystals strictly depends on the type of added stabilizers, while their diameter is also affected by the ice cream composition [53,89,90]. This result was also obtained for proteins binding free water in ice cream [94].

Based on the observations (Figure 4), it could be assumed that the mechanisms of retention of water molecules by the whey protein isolate added to the studied samples (NH3%, H3% and H5%) may be similar. Changes in the shape of the ice crystals, especially visible in NHC and NH3%, may indicate that recrystallization processes have occurred. In contrast, HC, H3% and H5% showed smaller crystal diameters, which moreover have regular shapes, compared to samples based on non-hydrolyzed concentrate (NHC and NH3%). A noticeable feature of NHC and NH3% is also a different appearance of the crystal structure, in which the crystals are arranged quite densely, and clearly defined edges create a three-dimensional effect. The shape of the crystals in NH3% indicates that coalescence and migration processes have taken place.

### 3.6. Quality Indicators

The overrun of ice cream for HC, H3% and H5% is the highest (71.98–79.18%) (Table 6), which is associated with the presence of monosaccharides (glucose, galactose) and their influence on the viscosity of ice cream mixes.

At the same time, even with the addition of whey protein isolate, the H3% and H5% have higher overrun than NHC and NH3%, although it is still slightly lower than HC. Lee and Duggan [60] reported on the foaming properties of whey protein isolate microgels. Due to the formation of elastic bonds, whey protein isolate contributes to the formation of more intense intermolecular interactions [95], as well as the uniform distribution of air bubbles, which ensure the production of ice cream with a high overrun. On the other hand, the addition of whey protein isolate, even in a smaller amount (3%), leads to a decrease in overrun, compared to the control sample (HC), because it increases the viscosity of the mixes, which limits the saturation of the mix with air during freezing [96]. Ice cream with non-hydrolyzed whey concentrates has a higher resistance to melting than samples based on hydrolyzed concentrates. The high content of monosaccharides, which are more effective cryoprotectants, leads to a decrease in the resistance to melting of HC, H3% and H5%. Whey protein isolate increases resistance to melting, but in samples with hydrolyzed concentrates, this indicator is still lower than in samples NHC and NH3%. A correlation was observed between the values of overrun and resistance to melting: the higher the air saturation of the ice cream, the higher the speed of its melting, respectively. This trend is comparable to the data of other scientists who determined overrun and resistance to melting in ice cream and frozen desserts [96,97,98]. However, Warren and Hartel [99] found that ice cream with a low overrun melts quickly, while ice cream with a high overrun slows down the melting rate.

Other scientists reported that there is no interdependence between the overrun and resistance to melting [54,100]. The speed of ice cream melting could also be affected by the size of ice crystals [54], and the larger they are, the lower the resistance to melting, which is consistent with our data on the size of ice crystals in experimental samples based on non-hydrolyzed whey concentrates (NHC, NH3%). The presence of hydrolysis products in the samples HC, H3% and H5% increases the melting rate, despite the formed small-size ice crystals in these samples. Lindamo et al. [101] reported a similar trend of decreasing melting resistance as the degree of lactose hydrolysis increased in ice cream samples. Due to its high moisture-binding capacity, whey protein isolate increases the viscosity of ice cream mixes [51], which limits the excessive growth of ice crystals, ensures the formation of a creamy consistency of ice cream and increases the resistance to melting of ice cream due to the structural and mechanical factor of foam stabilization as a dispersed system. For the purpose of a deeper study of the regularities of the formation of physicochemical parameters of ice cream, their hardness was determined. The hardness of HC with hydrolyzed concentrate (*p* < 0.05) is less than that of NHC, but the further addition of whey protein isolate to the composition of ice cream increases the hardness in samples H3% and H5%, although it is slightly less, compared to NH3%. During 14 days of storage, the hardness of all ice cream samples increased, and this was most noticeable for H3% and H5%. Patel, Baer and Acharya [102] reported that increasing the protein content of vanilla ice cream resulted in excessive firmness with lower overrun. The authors explained this by a hard three-dimensional gel formed by proteins, which led to an increase in viscosity. The obtained data coincide with the studied dynamics of crystal formation in ice cream, confirming that the larger the ice crystals, the higher the hardness of the ice cream, but this correlation is also not always confirmed by scientists [12,54]. Alfaifi and Stathopoulos [103] reported that the addition of whey protein isolate increased ice cream hardness, and Danesh et al. [104], on the contrary, found that hardness decreases when using whey protein isolate. From the data in Table 6, there is no direct dependence of the hardness on the resistance to melting or overrun, which is related to the peculiarities of the chemical composition of whey ice cream samples as a food system, in particular, the presence of lactose or monosaccharides, a reduced MSNF content and an increased protein content. The existing contradictions in the scientific literature regarding the interdependence of whipping, resistance to melting and hardness, as well as their correlation with the processes of ice crystal formation and saturation of ice cream with air bubbles are due to the difficulty of obtaining data on ice cream samples with an individual recipe composition. These dynamics may also depend on many factors and require a complex approach to analysis.

### 3.7. Microbiological Analysis

The number of *Lactobacillus acidophilus* probiotic cells for all samples was not lower than 6.2 log CFU/g during two weeks of storage (Table 7), which allows the developed ice cream compositions to be classified as probiotic, which means it contains at least 6 log CFU/g of probiotic cultures [105].

The decrease in *Lactobacillus acidophilus* from 6.6 log CFU/g (sample 1) to 6.2–6.3 log CFU/g (sample 2) is likely due to a high solid content, resulting in a decrease in water activity and an increase in osmotic pressure, which negatively affects the vital activity of starter microorganisms. The increase in the number of *Lactobacillus acidophilus* bacterial cells in samples with hydrolyzed whey concentrate, despite an even greater increase in the solid content, is due to the presence of glucose and galactose, which are a nutrient medium and stimulate the development process of probiotic cultures [42]. The presence of whey protein isolate could also contribute to the development of *Lactobacillus acidophilus* to some extent. Afzaal et al. [106] showed that whey protein isolates were more effective due to their amino acid composition as a protective medium for probiotic cell strains. Burgain et al. [107] reported that some of the molecules present in the cells of probiotic bacteria are involved in adhesion with polysaccharides, acids, proteins and lipids. Milk protein ingredients, as representatives of biopolymers, are common components of bioactive agents (encapsulants, protectors) used to protect probiotic bacteria [108,109]. The combination of probiotics with whey protein isolate can add more value to processed foods, but the main disadvantage associated with this combination is the instability of bacteria, since WPI is an ideal food source for the growth and reproduction of microorganisms at high moisture content [40]. The results of counting yeasts and fungi in the ice cream samples indicate that H3% and H5% have a lower content of them compared to other samples. Due to the active binding of free water and increased osmotic pressure in the product, whey protein isolate creates unfavorable conditions for the development of fungi and yeast, which correlates with the obtained data on the activity of water in ice cream mixes.

## 4. Conclusions

Whey protein isolate has a significant effect on the rheological properties of ice cream mixes based on hydrolyzed whey concentrates, namely, an increase in viscosity indicators. Substantial freezing of free water in ice cream occurs in the temperature range from the cryoscopic temperature to −10 °C, which ensures the freezing of 70.5–71.2% of water and, in samples HC, H3% and H5% up to 72.9–76.1%. Whey protein isolate contributes to the formation of a structure with increased water-holding capacity, which leads to the formation of a more uniform crystal structure (average diameter of ice crystals from 13.75 to 14.75 μm) in HC, H3% and H5% samples. Analysis of the microstructure confirms the feasibility of using whey protein isolate (3–5%), which ensures an even distribution of air bubbles in the ice cream and contributes to obtaining a product with a high whipping index. The combination of non-hydrolyzed whey concentrate and 3% whey protein isolate (NH3%) provides the greatest stability to preserve the purity and color intensity of the ice cream during storage. According to the microbiological analysis, the developed types of ice cream can be classified as probiotic based on the amount of Lactobacillus acidophilus in all samples not being lower than 6.2 log CFU/g. According to the results of the research, H3% and H5% were selected as the best ice cream recipes, which ensure the proper formation of quality indicators of whey ice cream. The prospect of further research is the study of a complex of quality indicators for acidophilic ice cream based on hydrolyzed whey concentrates during storage.

## Figures and Tables

**Figure 1 foods-13-00170-f001:**
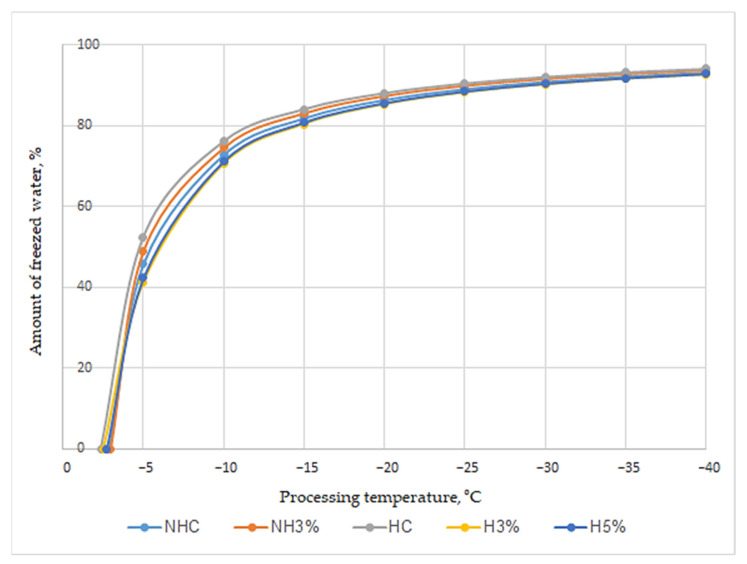
Free water freezing dynamics under different modes of low-temperature processing. **Note.** NHC—ice cream based on non-hydrolyzed whey concentrate; NH3%—ice cream based on non-hydrolyzed whey concentrate + 3% whey protein isolate; HC—ice cream based on hydrolyzed whey concentrate; H3%—ice cream based on hydrolyzed whey concentrate + 3% whey protein isolate; H5%—ice cream based on hydrolyzed whey concentrate + 5% whey protein isolate.

**Figure 2 foods-13-00170-f002:**
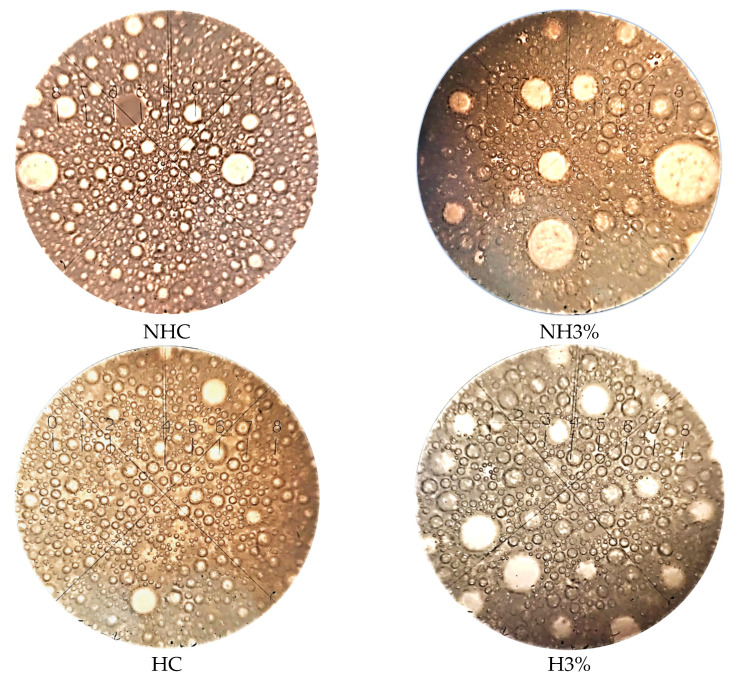
Microstructure of soft ice cream at a magnification of 160 times. NHC—ice cream based on non-hydrolyzed whey concentrate; NH3%—ice cream based on non-hydrolyzed whey concentrate + 3% whey protein isolate; HC—ice cream based on hydrolyzed whey concentrate; H3%—ice cream based on hydrolyzed whey concentrate + 3% whey protein isolate; H5%—ice cream based on hydrolyzed whey concentrate + 5% whey protein isolate.

**Figure 3 foods-13-00170-f003:**
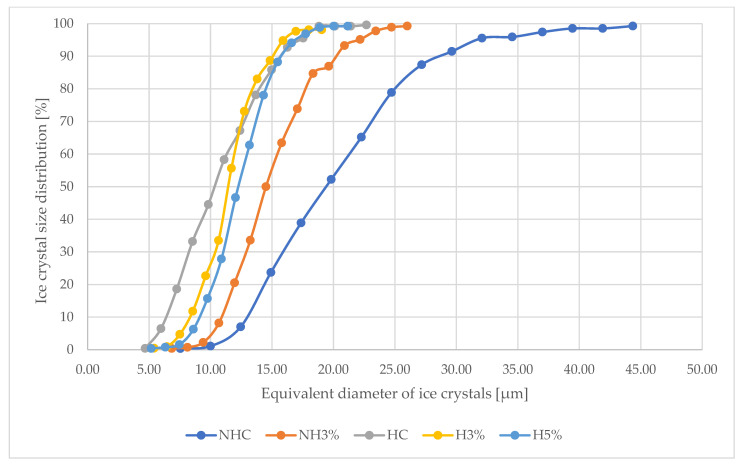
Distribution of ice crystals in whey ice cream samples. **Note.** NHC—ice cream based on non-hydrolyzed whey concentrate; NH3%—ice cream based on non-hydrolyzed whey concentrate + 3% whey protein isolate; HC—ice cream based on hydrolyzed whey concentrate; H3%—ice cream based on hydrolyzed whey concentrate + 3% whey protein isolate; H5%—ice cream based on hydrolyzed whey concentrate + 5% whey protein isolate.

**Figure 4 foods-13-00170-f004:**
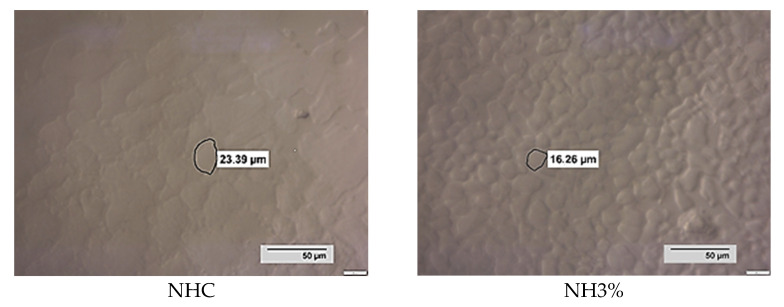
Photographs of ice crystals after 24 h of storage at −18 °C. Note. NHC—ice cream based on non-hydrolyzed whey concentrate; NH3%—ice cream based on non-hydrolyzed whey concentrate + 3% whey protein isolate; HC—ice cream based on hydrolyzed whey concentrate; H3%—ice cream based on hydrolyzed whey concentrate + 3% whey protein isolate; H5%—ice cream based on hydrolyzed whey concentrate + 5% whey protein isolate.

**Table 1 foods-13-00170-t001:** Formulations of experimental samples of whey ice cream.

Ingredients, %	Ice Cream Samples				
	NHC	NH3%	HC	H3%	H5%
Non-hydrolyzed concentrate of demineralized whey	75.0	75.0	–	–	–
Hydrolyzed concentrate of demineralized whey	–	–	75.0	75.0	75.0
White sugar	11.0	11.0	9.0	9.0	9.0
Whey protein isolate 90%	–	3.0	–	3.0	5.0
Stabilization system	0.6	0.6	0.6	0.6	0.6
Activated starter	3.0	3.0	3.0	3.0	3.0
Vanilla	0.1	0.1	0.1	0.1	0.1
Water	10.3	7.3	12.3	9.3	7.3
Total	100.0	100.0	100.0	100.0	100.0

**Note.** NHC—ice cream based on non-hydrolyzed whey concentrate; NH3%—ice cream based on non-hydrolyzed whey concentrate + 3% whey protein isolate; HC—ice cream based on hydrolyzed whey concentrate; H3%—ice cream based on hydrolyzed whey concentrate + 3% whey protein isolate; H5%—ice cream based on hydrolyzed whey concentrate + 5% whey protein isolate.

**Table 2 foods-13-00170-t002:** Chemical composition of whey concentrates and whey ice cream.

Sample	Total Solids, %	Protein, %	Fat, %	Carbohydrates, %	Lactose, %	The Degree of Lactose Hydrolysis, %	Level of Protein Supply, %	Ice Cream Category/ Nutritional Status
Whey concentrates								
NHCDW	39.92 ^a^ ± 0.30	4.45 ^a^ ± 0.12	0.43 ^b^ ± 0.01	30.71 ^a^ ± 0.68	30.50 ^b^ ± 1.19	–	–	–
HCDW	40.01 ^a^ ± 0.84	4.41 ^a^ ± 0.55	0.40 ^a^ ± 0.01	30.61 ^a^ ± 0.51	1.29 ^a^ ± 0.04	98.71 ^a^ ± 0.04	–	–
Whey ice cream								
NHC	41.61 ^ab^ ± 1.24	3.31 ^a^ ± 0.11	0.74 ^abc^ ± 0.02	33.05 ^a^ ± 1.52	22.90 ^b^ ± 0.53	–	8.92 ^a^ ± 0.52	premium
NH3%	44.55 ^b^ ± 0.87	6.10 ^b^ ± 0.25	0.72 ^a^ ± 0.01	33.46 ^a^ ± 0.79	22.87 ^b^ ± 0.12	–	15.11 ^b^ ± 0.35	super premium, enriched with protein
HC	39.58 ^a^ ± 1.53	3.30 ^a^ ± 0.14	0.72 ^a^ ± 0.03	33.10 ^a^ ± 0.65	0.97 ^a^ ± 0.01	99.03 ^a^ ± 0.01	8.89 ^a^ ± 0.27	super premium
H3%	42.54 ^ab^ ± 1.14	6.02 ^b^ ± 0.11	0.78 ^bc^ ± 0.01	33.28 ^a^ ± 0.47	0.98 ^a^ ± 0.03	99.02 ^a^ ± 0.03	15.02 ^b^ ± 0.19	super premium, enriched with protein
H5%	44.63 ^b^ ± 1.01	7.84 ^c^ ± 0.05	0.79 ^b^ ± 0.01	33.53 ^a^ ± 0.34	0.98 ^a^ ± 0.01	99.02 ^a^ ± 0.01	18.59 ^c^ ± 0.13	super premium, enriched with protein

**Note.** NHCDW—non-hydrolyzed concentrate of demineralized whey; HCDW—hydrolyzed concentrate of demineralized whey; NHC—ice cream based on non-hydrolyzed whey concentrate; NH3%—ice cream based on non-hydrolyzed whey concentrate + 3% whey protein isolate; HC—ice cream based on hydrolyzed whey concentrate; H3%—ice cream based on hydrolyzed whey concentrate + 3% whey protein isolate; H5%—ice cream based on hydrolyzed whey concentrate + 5% whey protein isolate. ^a–c^—different superscript letters in the columns represent significant differences in the mean values of the same parameter (*p* ≤ 0.05).

**Table 3 foods-13-00170-t003:** Thermophysical characteristics of whey ice cream mixes.

Indicator		Sample				
		NHC	NH3%	HC	H3%	H5%
Foam overrun, %	5	144.58 ^a^ ± 1.67	158.74 ^c^ ± 2.10	150.08 ^b^ ± 1.5	203.33 ^d^ ± 1.7	192.14 ^e^ ± 2.0
	10	177.51 ^a^ ± 1.59	184.96 ^a^ ± 1.47	201.42 ^b^ ± 4.2	236.78 ^d^ ± 2.5	220.43 ^c^ ± 3.2
	15	164.23 ^a^ ± 2.54	179.7 ^b^ ± 1.89	187.54 ^c^ ± 1.6	246.75 ^e^ ± 4.7	221.59 ^d^ ± 1.2
Foam resistance, хв	5	31.57 ^a^ ± 0.87	33.89 ^b^ ± 0.27	44.82 ^c^ ± 0.67	48.41 ^d^ ± 0.74	45.90 ^c^ ± 0.81
	10	32.46 ^a^ ± 0.54	39.94 ^b^ ± 0.71	49.55 ^c^ ± 0.56	58.45 ^d^ ± 0.25	57.63 ^d^ ± 0.90
	15	32.11 ^a^ ± 0.68	38.43 ^b^ ± 0.44	48.13 ^c^ ± 1.36	61.94 ^e^ ± 0.49	59.33 ^d^ ± 1.01
Viscosity, Mpa × s		254.42 ^b^ ± 1.86	384.55 ^d^ ± 1.12	228.61 ^a^ ± 2.54	339.47 ^c^ ± 1.07	511.05 ^e^ ± 1.24
Cryoscopic temperature, °C		−2.88 ^a^ ± 0.01	−2.95 ^a^ ± 0.08	−2.39 ^d^ ± 0.01	−2.55 ^c^ ± 0.02	−2.71 ^b^ ± 0.05
Water activity, units		0.955 ^e^ ± 0.02	0.941 ^d^ ± 0.03	0.912 ^c^ ± 0.01	0.905 ^b^ ± 0.01	0.896 ^a^ ± 0.02

**Note.** NHC—ice cream based on non-hydrolyzed whey concentrate; NH3%—ice cream based on non-hydrolyzed whey concentrate + 3% whey protein isolate; HC—ice cream based on hydrolyzed whey concentrate; H3%—ice cream based on hydrolyzed whey concentrate + 3% whey protein isolate; H5%—ice cream based on hydrolyzed whey concentrate + 5% whey protein isolate. 5, 10, 15—time of ice cream mix whipping, min. ^a–e^—different superscript letters in the lines represent significant differences in the mean values of the same parameter (*p* ≤ 0.05).

**Table 4 foods-13-00170-t004:** Color parameters of ice cream mixes.

Sample			Color Parameters				
			L*	a*	b*	C*	h°
NHC	mix		72.80 ^a^ ± 1.08	−4.38 ^b^ ± 0.12	20.50 ^a^ ± 0.88	20.96 ^a^ ± 0.66	102.07 ^a^ ± 2.28
	ice cream	1st day	81.66 ^a^ ± 2.54	−4.09 ^b^ ± 0.04	20.16 ^a^ ± 0.50	14.73 ^a^ ± 0.05	101.50 ^a^ ± 2.05
		14th day	76.81 ^a^ ± 1.27	−4.11 ^b^ ± 0.17	21.84 ^ab^ ± 0.17	13.52 ^a^ ± 0.64	103.17 ^b^ ± 1.84
NH3%	mix		69.56 ^a^ ± 1.59	−3.61 ^a^ ± 0.09	20.18 ^a^ ± 0.01	20.37 ^a^ ± 0.95	100.24 ^a^ ± 1.10
	ice cream	1st day	82.78 ^a^ ± 2.22	−2.98 ^a^ ± 0.13	21.03 ^a^ ± 0.67	21.26 ^b^ ± 0.54	98.08 ^a^ ± 2.89
		14th day	76.07 ^a^ ± 2.18	−3.05 ^a^ ± 0.01	21.56 ^ab^ ± 0.94	20.90 ^b^ ± 0.96	98.99 ^a^ ± 2.17
HC	mix		79.03 ^a^ ± 2.01	−3.94 ^a^ ± 0.11	24.07 ^a^ ± 0.58	24.41 ^a^ ± 1.14	99.19 ^a^ ± 2.74
	ice cream	1st day	82.31 ^a^ ± 0.87	−2.42 ^a^ ± 0.10	14.55 ^b^ ± 0.69	20.57 ^a^ ± 0.71	99.11 ^a^ ± 2.54
		14th day	80.98 ^a^ ± 1.85	−2.55 ^a^ ± 0.02	14.69 ^a^ ± 0.54	18.09 ^b^ ± 0.05	99.54 ^a^ ± 1.71
H3%	mix		81.36 ^a^ ± 1.54	−3.17 ^a^ ± 0.13	24.43 ^a^ ± 0.47	24.66 ^a^ ± 1.19	97.46 ^a^ ± 0.86
	ice cream	1st day	83.25 ^a^ ± 1.73	−2.84 ^a^ ± 0.08	18.84 ^a^ ± 0.54	18.57 ^a^ ± 0.55	98.23 ^a^ ± 3.57
		14th day	78.56 ^a^ ± 1.25	−2.91 ^a^ ± 0.11	19.63 ^ab^ ± 0.07	18.08 ^b^ ± 0.76	99.07 ^a^ ± 2.60
H5%	mix		77.20 ^a^ ± 1.89	−2.71 ^a^ ± 0.01	23.39 ^a^ ± 1.05	23.08 ^a^ ± 1.00	96.54 ^a^ ± 0.97
	ice cream	1st day	80.01 ^a^ ± 2.12	−3.30 ^a^ ± 0.13	21.58 ^a^ ± 0.68	21.84 ^a^ ± 0.58	98.33 ^a^ ± 1.18
		14th day	77.88 ^a^ ± 2.57	−3.35 ^a^ ± 0.05	22.42 ^b^ ± 0.84	20.05 ^b^ ± 1.13	98.74 ^a^ ± 1.36

**Note.** NHC—ice cream based on non-hydrolyzed whey concentrate; NH3%—ice cream based on non-hydrolyzed whey concentrate + 3% whey protein isolate; HC—ice cream based on hydrolyzed whey concentrate; H3%—ice cream based on hydrolyzed whey concentrate + 3% whey protein isolate; H5%—ice cream based on hydrolyzed whey concentrate + 5% whey protein isolate. ^a,b^—different superscript letters in the columns represent significant differences in the mean values of the same parameter (*p* ≤ 0.05).

**Table 5 foods-13-00170-t005:** Comparison of ice crystal sizes after 24 h of storage at −18°C.

Sample	Minimum Diameter of Ice Crystals (μm)	Maximum Diameter of Ice Crystals (μm)	The Average Value of the Diameter of Ice Crystals (μm)
NHC	7.55 ^d^ ± 0.12	44.36 ^d^ ± 2.03	25.96 ^c^ ± 1.04
NH3%	6.85 ^c^ ± 0.09	26.00 ^c^ ± 0.52	16.43 ^b^ ± 1.17
HC	4.68 ^a^ ± 0.10	22.68 ^b^ ± 0.16	13.68 ^a^ ± 0.02
H3%	5.41 ^b^ ± 0.24	19.04 ^a^ ± 0.12	12.23 ^a^ ± 0.18
H5%	5.18 ^a^ ± 0.02	21.17 ^ab^ ± 0.95	13.18 ^a^ ± 0.56

**Note.** NHC—ice cream based on non-hydrolyzed whey concentrate; NH3%—ice cream based on non-hydrolyzed whey concentrate + 3% whey protein isolate; HC—ice cream based on hydrolyzed whey concentrate; H3%—ice cream based on hydrolyzed whey concentrate + 3% whey protein isolate; H5%—ice cream based on hydrolyzed whey concentrate + 5% whey protein isolate. ^a–d^—different superscript letters in the columns represent significant differences in the mean values of the same parameter (*p* ≤ 0.05).

**Table 6 foods-13-00170-t006:** Physicochemical indicators of ice cream samples.

Indicator			Ice Cream Samples				
			NHC	NH3%	HC	H3%	H5%
Overrun, %			71.84 ^b^ ± 1.45	59.3 ^a^ ± 0.86	79.18 ^c^ ± 2.55	76.55 ^bc^ ± 3.08	71.98 ^b^ ± 2.72
pH		1st day	5.25 ^a^ ± 0.01	5.22 ^a^ ± 0.05	5.23 ^a^ ± 0.08	5.19 ^a^ ± 0.03	5.17 ^a^ ± 0.01
		14th day	5.20 ^b^ ± 0.10	5.13 ^ab^ ± 0.04	5.12 ^ab^ ± 0.01	5.09 ^ab^ ± 0.01	5.05 ^a^ ± 0.04
Resistance to melting	1 drop	1st day	29.81 ^b^ ± 0.53	34.58 ^c^ ± 1.04	24.85 ^a^ ± 0.95	26.07 ^a^ ± 1.17	29.11 ^b^ ± 1.50
		14th day	30.11 ^bc^ ± 1.08	35.87 ^d^ ± 1.55	25.04 ^a^ ± 0.37	28.54 ^b^ ± 1.20	32.42 ^c^ ± 1.16
	10 cm^3^	1st day	44.08 ^b^ ± 1.62	49.76 ^c^ ± 1.89	33.69 ^a^ ± 1.36	36.25 ^a^ ± 1.24	40.67 ^b^ ± 1.33
		14 th day	46.11 ^c^ ± 1.91	52.32 ^d^ ± 1.01	34.78 ^a^ ± 0.88	39.82 ^b^ ± 1.45	43.83 ^c^ ± 1.57
Hardness, g/cm^3^	1st day		1734.88 ^b^ ± 44.37	2512.46 ^d^ ± 50.27	1580.27 ^a^ ± 41.86	1941.43 ^c^ ± 51.87	2409.74 ^d^ ± 47.08
	14th day		1808.51 ^b^ ± 30.27	2567.09 ^d^ ± 14.90	1602.82 ^a^ ± 55.68	2154.37 ^c^ ± 44.20	2618.74 ^d^ ± 36.17

Note. NHC—ice cream based on non-hydrolyzed whey concentrate; NH3%—ice cream based on non-hydrolyzed whey concentrate + 3% whey protein isolate; HC—ice cream based on hydrolyzed whey concentrate; H3%—ice cream based on hydrolyzed whey concentrate + 3% whey protein isolate; H5%—ice cream based on hydrolyzed whey concentrate + 5% whey protein isolate. ^a–d^—different superscript letters in the lines represent significant differences in the mean values of the same parameter (*p* ≤ 0.05).

**Table 7 foods-13-00170-t007:** Microbiological indicators of ice cream samples.

Sample	Storage Time (Days)	Counting (log CFU/g)			
		Coliform	LA-5^®^	Yeasts	Fungi
**NHC**	1	ND	6.6	5.1	6.0
	14	ND	6.6	5.5	6.2
**NH3%**	1	ND	6.3	5.4	ND
	14	ND	6.2	5.7	5.1
**HC**	1	ND	>7.7	5.3	5.3
	14	ND	>7.7	5.5	5.5
**H3%**	1	ND	7.6	5.0	ND
	14	ND	7.6	5.3	ND
**H5%**	1	ND	7.4	5.0	ND
	14	ND	7.3	5.2	ND

Note. ND—not detected. NHC—ice cream based on non-hydrolyzed whey concentrate; NH3%—ice cream based on non-hydrolyzed whey concentrate + 3% whey protein isolate; HC—ice cream based on hydrolyzed whey concentrate; H3%—ice cream based on hydrolyzed whey concentrate + 3% whey protein isolate; H5%—ice cream based on hydrolyzed whey concentrate + 5% whey protein isolate.

## Data Availability

Data is contained within the article.

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
