# Peer review of "The Influence of Whey Protein Isolate on the Quality Indicators of Acidophilic Ice Cream Based on Liquid Concentrates of Demineralized Whey"

_foods, 2024, doi:10.3390/foods13010170_

Round 1

Reviewer 1 Report

Comments and Suggestions for Authors

The paper is well written and the study is considered novel because the hydrolysed whey concentrate based ice cream containing probiotics was prepared and the effect of whey protein isolate on the quality and thermal behaviour of ice cream was investigated, which is important research for the preparation of functional food product. The experimental design and analyses were found to be compatible and appropriate to the topic of the article. The manuscript needs improvement inthe suggested following  issues.

Line 97:Please add related citation.

Line 116: which sugar is used?

Table 3 “different superscript letters in the columns represent significant” is it line ? Viscosity differences should be in the line?

For foam analyses, two-way comparisons are required for both rows and columns.

Lin 334: It is not necessary for the discussion to mention the effect of Sucrose as an unused ingredient.

Line 480: MSNF?

Line 480: air has a low heat transfer rate?

Line 491: Why did the authors choose to correlate the smaller ice crystal size with longer storage stability without recrystallisation instead of reporting the observation of ice cream samples stored for 14 days? Why were the results of the 14-day samples not reported along with the 1-day results?

Line 564: Same line instead of same column for overrun? And please, revise the letters other results.

Line 570 Please add discussion why adding 3% whey isolate decreased the overrun values dramatically?

Line 583: The melting resistance results of the study are lower for the samples with smaller ice crystal sizes. Therefore, this sentence does not support your results. Please revise this sentence.

Please discuss: why did the samples with lower ice crystals and higher viscosity and overrun values have lower melt resistance?

Please discuss time (1 and 14. Day) effect on hardness value

Line 624: viscosity of HC samples were not found in lower viscosities than untreated samples.

Reviewer 2 Report

Comments and Suggestions for Authors

The authors of this study aimed to examine the effect of added whey protein isolate to ice creams mixes based on hydrolyzed and non-hydrolyzed liquid concentrates of demineralized whey on their rheological and thermophysical properties, as well as color, microstructure, microscopy, microbiological and quality analysis.  The paper is well structured, with adequate discussion and appropriate references,  and I noticed only some minor flaws listed below:

Line 115-135, Write by which technologies the liquid concentrates of demineralized whey and whey protein isolate were produced.

Line 144-146, Describe in more detail by which technique the hydrolyzed concentrates were obtained and by which enzymes.

Line 302-318, It would be useful to provide a table in which the categories of ice cream types are presented depending on the content of chemical components.

Line 118-151, Compare the obtained color parameters with other authors who examined similar  recipes for whey ice cream mixes.

Line 180, MSFN, Write the full meaning of the abbreviation.

In the conclusion, highlight the practical importance of the work, emphasize which ice cream formulation is recommended or if additional research is needed.

Author Response

First of all, we would like to thank you for your attention to our article and the opportunity to improve the quality of the article. According to the comments, the following corrections were made:

Line 115-135, Write by which technologies the liquid concentrates of demineralized whey and whey protein isolate were produced.

The information was added:

‘Liquid concentrates of demineralized whey with solids content of 40% were used to make experimental ice cream samples. Non-hydrolyzed concentrates of demineral-ized whey were produced by reconstitution of whey powder with degree of deminer-alization of 90% (Herkules, MLEKOVITA, Poland) in water. Hydrolyzed concentrates of demineralized whey were obtained by lactose fermenting using the enzyme lactase (β-galactosidase) with an activity of 5200 NLU/g (SEROWAR s.c., Poland) and the starter preparation nu-trish® LA-5® containing L. acidophilus (Chr. Hansen A/S, Denmark). Whey protein isolate 90% (SPOMLEK, Poland) was used as a protein sup-plement. Water, white (regular) sugar, vanillin, activated starter and the stabilization system Cremodan SI 320 (Danisco A/S, Denmark) were used to prepare ice cream mix-es.’ (lines 116-125)

Line 144-146, Describe in more detail by which technique the hydrolyzed concentrates were obtained and by which enzymes.

The information was added:

‘Demineralized whey powder was reconstituted in water at a temperature of (40-45) °C to obtain concentrates with solids content of 40%. The concentrates were filtered, pasteurized at (85-88) °C for 3-5 minutes and cooled to the storage tempera-ture for non-hydrolyzed concentrates.

Hydrolyzed concentrates of demineralized whey were obtained according to the technology of Osmak et al. [42]. For the production of hydrolyzed whey concentrates, after pasteurization, they were cooled to (40-43) °C and simultaneously fermented with β-galactosidase preparation and starter nu-trish® LA-5®. The simultaneous in-troduction of the β-galactosidase enzyme and the starter during the lag phase of L. ac-idophilus development (2-4 hours), the enzyme has time to reveal hydrolytic activity at pH ≥ 5.7, which makes it possible to achieve maximum hydrolysis of lactose within 8 hours of enzymolysis.’ (lines 149-160)

Line 302-318, It would be useful to provide a table in which the categories of ice cream types are presented depending on the content of chemical components.

The information about ice cream category and nutritional status was added to Table 2 (line 313).

Line 118-151, Compare the obtained color parameters with other authors who examined similar  recipes for whey ice cream mixes.

The discussion was added:

‘Barros et al. [76] also reported a decrease in whiteness in ice cream samples from 87.65 to 82.45 units depending on the content of concentrated whey (70–280 g per ice cream sample).’ (lines 434-436)

‘Barros et al. [76] also reported an increase in the b* indicator from 16.27 to 18.02% with an increase in the content of concentrated whey in the composition of ice cream. However, Meneses et al. [81] showed that the b* indicator increased from 57.64 to 60.92 units when adding whey at different mass fractions. A significant difference with the results obtained in this study may be related to the hydrolysis of lactose, as well as the type of whey used.’ (lines 458-463)

Line 180, MSFN, Write the full meaning of the abbreviation.

An explanation was added ‘MSNF (milk solids non-fat)’ (line 502).

In the conclusion, highlight the practical importance of the work, emphasize which ice cream formulation is recommended or if additional research is needed.

The information was added:

‘According to the results of the research, H3% and H5% were selected as the best ice cream recipes, which ensure the proper formation of quality indicators of whey ice cream. The prospect of further research is the study of a complex of quality indicators for acidophilic ice cream based on hydrolyzed whey concentrates during storage.’ (lines 699-703)

Round 2

Reviewer 1 Report

Comments and Suggestions for Authors

The authors were carefully revised and corrected the manuscript. I think it is now acceptable for publication